

# Derivation of land surface temperature from Landsat Thematic Mapper (TM) sensor data and analyzing relation between land use changes and surface temperature

S. Zareie[1], H. Khosravi[*2], A. Nasiri[3]

[1] Ph.D. student of GIS & Remote Sensing, Institute of Earth Sciences, Saint Petersburg State University, Saint Petersburg, Russian federation
[2] Department of Arid and Mountainous Regions Reclamation, Faculty of Natural Resources, College of Agriculture & Natural Resources, University of Tehran, Karaj, Iran (hakhosravi@ut.ac.ir)
[3] Department of Ecology and Environmental Management, Protection of the Natural Resource and Environment, Land Cadastre Faculty, State University of Land Use Planning, Moscow, Russia

*Correspondence to*: h. khosravi (hakhosravi@ut.ac.ir)

*Abstract*

Land surface temperature (LST) is one of the key parameters in the physics of land surface processes from local to global scales, and it is one of the indicators of environmental quality. Evaluation of the surface temperature distribution and its relation with existing land use types are very important to investigate the urban microclimate. Land use planning in the cities must be in accordance with sustainable development goals to make the urban environment without soil, water and air pollution. In the arid and semi-arid regions, understanding the role of land use changes in the formation of urban heat islands is necessary to provide urban planning to control or reduce surface temperature. Internal factors and environmental conditions of the Yazd city have important role in the formation of special thermal conditions in the Iran. In this paper, we used the Temperature Emissivity Separation (TES) algorithm for LST retrieving from TIRS data of the Landsat Thematic Mapper (TM). The Root Mean-Square Error (RMSE) and Coefficient of Determination (R2) were used for validation of retrieved LST values. Land use types of the Yazd city were identified and relations between land use types, land surface temperature and NDVI index were analyzed. The Kappa coefficient and overall accuracy were calculated for accuracy assessment of the land use classification. The results of this study showed that optical and thermal remote sensing methodologies can be used to research urban environmental parameters. Finally, it was found that special thermal conditions in Yazd city were formed by land use changes. Increasing the area of asphalt roads, residential, commercial and industrial types and decreasing the area of the parks, green spaces and fallow lands types of land uses in the Yazd city caused a rise in surface temperature during the 11-year period.

**Keywords:** Yazd, NDVI, Landsat Thematic Mapper, LST, Land use.

## 1 Introduction

The main problem in the urban areas is surface temperature increasing due to conversion of vegetated surfaces into asphalt roads, residential, commercial and industrial areas. Nowadays, climate change in the cities is occurring by anthropogenic activities and land use changes. The atmospheric conditions of the urban areas, land surface temperature, warming, evaporation and absorption of solar radiation may be changed by anthropogenic changes. The study of surface temperature in the cities that located in the arid and semi-arid areas is necessary, because high temperature leads to energy consumption during buildings cooling, which is



economically very costly especially in the warm months of year. Remote sensing instruments are key players to study and mapping land surface temperature (LST) at temporal and spatial scales (André et al., 2015). LST indicator shows effects of different types of phenomena and features in the electromagnetic energy dispatch (Bingwei Tian et al., 2015). Remote sensing methodology requires less time and lower cost than field methods to investigate various phenomena on the land surface (Niu et al., 2015). The

advantages of using remote sensing methodology are: the repetitive and consistent coverage, high resolution and evaluation of land surface characteristics (Owen et al., 1998). Thermal infrared (TIR) data in the remote sensing can help us obtain quantitative information of surface temperature. Landsat imagery can be applied for monitoring different types of land use in arid and semi-arid regions (Baojuan et al., 2015). The Landsat TM and ETM+ sensors images can be used to study relation between surface temperature and land use types using thermal quantitative indicators (Weng 2003, Streutker, 2003). Land surface temperature can

be retrieved using data from NOAA-11 AVHRR channels 4 and 5 by emissivity calculation (France et al., 1994). The using of LST values which vary according to the surface characteristics is a new method for investigating the effects of land surface different features on the surface temperature especially in urban areas (Guanhua et al., 2015). In the several studies the relative warmth of cities was estimated by knowing air temperature and land use changes. LST index provide important information about climate and the surface physical characteristics. Land use changes and anthropogenic activities are affecting the environment and land

surface temperature (Dehua et al., 2012; Weng and Schubring, 2004). The estimation of the LST from the radiative transfer equation, the mono-window and single-channel algorithms can be used to retrieve the land surface temperature (LST) from thermal infrared data of the Thematic Mapper (TM) sensor (José et al., 2004). The Normalized Difference Vegetation Index (NDVI) is a good indicator for identifying long term changes in the vegetation covers and their status (Baihua and Isabela, 2015). The NDVI calculation method using surface emissivity can be applied to areas with different soil and vegetation types and where the

vegetation cover changes (Valor et al., 1996). Therefore, analysis of spatial variability of NDVI, surface temperature and relation between these parameters is essential in the environmental studies. Combined study of NDVI, surface temperature and temporal relation these two parameters with land use changes can be used for investigating climate change and global warming (Schultz and Halpert, 1993). Vegetation cover change is the main factor which causes surface temperature changes. It should be noted that increasing surface temperature may increase vegetation cover density in the area, of course in areas where there are sufficient water

resources (Weixin et al., 2011). Different types of vegetation cover have different spatial responses to climate changes (Dehua et al., 2012). In the environmental studies, researchers have investigated land surface temperature using vegetation indices (Wei et al., 2015). Analysis of NDVI and LST of the different times (days, months, seasons and years) can be used to detect land use changes, which were formed because of deforestation, forest fires, mining activities, urban expansion and grassland regeneration (Sandra et al., 2015). Changes in land use and land cover can be evaluated by analysis of the vegetation cover and NDVI trends.

The vegetation phenology was detected using Terra MODIS NDVI data by Gong Z. et al. (2015). Derivation of land surface temperature (LST) from medium to high spatial resolution data of remote sensing is very important to study climate change and environment (Juan et al., 2014). Land use changes, vegetation cover and soil moisture have strong effects on the land surface temperature; therefore, surface temperature can be applied to study land use changes, urbanization and desertification. Surface emissivity calculation is important to estimate surface temperature. In the several studies, laboratory measurements of the

emissivity data were used for estimation of land surface temperature (Salisbury and D'Aria, 1992; Salisbury and D'Aria, 1994). In the present study, heterogeneous surface temperature and NDVI index of Yazd city in the Iran were calculated using Landsat TM sensor data. Surface temperature variation over different land use types in the Yazd city are investigated, and analyzed the relation between NDVI index and land surface temperature. The aims of this study are to apply Temperature Emissivity Separation algorithm (TES) for LST retrieving from Landsat TM thermal data and to analyze the NDVI index, different land use types and




their roles in the surface temperature change. The main advantages of this study are TES algorithm (calculating emissivity) for retrieving LST, full statistical analysis for results validation and simultaneous analysis of NDVI, LST and land use.

## 2 Material and methods

### 2.1 Study area

Yazd city has been chosen as the study area, since there are combination of different land use categories and susceptible to dust storms. Yazd is located in 31˚47'37"-31˚57'56" north latitude and 54˚13'28"-54˚27'10" east longitude in the Iran. This city has an altitude of 1230 m and covering an area of 2,491 km2 (Figure 1). Study area is located in the arid and semi-arid belt in the northern hemisphere. In the arid and semi-arid areas, vegetation covers are affected by high diurnal and seasonal variations of temperature, low amount of precipitations and high evaporation (Dehghan, 2011). The predominant features of the territory are residential area,

waste land and bare soil. Land surface temperature in the Yazd city is affected by warm, arid and semi-arid climate, low precipitation and remoteness of major water resources such as Caspian Sea, Persian Gulf and Oman Sea.

Figure 1

### 2.2 Satellite data processing and methodology

### 2.2.1 Land surface emissivity (LSE) calculation:

Landsat TM sensor data of 08[th] Aug, 1998 and 06[th] Aug, 2009 (day time) were used in present study. Surface emissivity calculation is the first step of land surface temperature retrieving by Temperature Emissivity Separation (TES) algorithm. The emissivity per pixel was obtained directly from Landsat TM sensor data. Natural surfaces at the pixel scale (pixel resolution of 30 meters) are heterogeneous and they differ from each other in their emissivity. In addition, the surface emissivity is affected by surface roughness, vegetation cover and different land use types. In the present study, surface emissivity was evaluated by analysis of

NDVI index and the fraction of vegetation cover per pixel. Emissivity is a quantification of the intrinsic ability of a surface in converting heat energy into above surface radiation and depends on the physical properties of the surface and on observation conditions (Sobrino et. al. 2001). Surface emissivity can be extracted using NDVI values of the bare soil, fully vegetated and mixture of bare soil and vegetation (Sobrino et. al. 2004). In this study, following equation was used to extract land surface emissivity for each pixel:

$$\varepsilon = \varepsilon v * Pv + \varepsilon s * (1 - Pv) + d\varepsilon \qquad (1)$$

Where, $\varepsilon$ is the LSE, $Pv$ is the proportion of vegetation, $\varepsilon v$ is vegetation emissivity (0.99) and $\varepsilon s$ is soil emissivity (0.97). The term $d\varepsilon$ show geometric distribution effect of natural surfaces and their internal reflection. This term ($d\varepsilon$) for our study area was not considered because it is negligible for surfaces with little height difference. The proportion of vegetation ($Pv$) is calculated by following equation:

$$Pv = \left( \frac{NDVI - NDVI_{min}}{NDVI_{max} - NDVI_{min}} \right)^2 \qquad (2)$$

NDVI index is obtained from spectral reflectance measurements in the visible (RED) and near-infrared regions (NIR) by:

$$NDVI = \frac{NIR - RED}{NIR + RED} \qquad (3)$$

Theoretically, NDVI values of a given pixel scales vary between -1 and +1. The high values of NDVI indicate vegetation density and health. This method needs elementary knowledge of emissivity and NDVI of the different features and land use types.



### 2.2.2 Conversion calibrated Digital Numbers to spectral radiance (Qcal-to-$L_\lambda$):

The at-sensor spectral radiance is the amount of energy received by the satellite sensor. Calculation of spectral radiance is the fundamental step in converting satellite image data into a physically radiometric scale. Radiometric calibration of the Landsat TM sensor involves rescaling the raw digital numbers of the satellite image to calibrated digital numbers. The pixel values of unprocessed image data were converted to spectral radiance by radiometric calibration.

Spectral Radiance ($L_\lambda$) at the sensor's aperture in watts/(meter squared*ster*μm) is provided by following equation:

$$L_\lambda = Grescale * QCAL + Brescale \qquad (4)$$

Spectral Radiance is also expressed as:

$$L_\lambda = \left(\frac{LMAX-LMIN}{QCALMAX-QCALMIN}\right) * (QCAL - QCALMIN) + LMIN \qquad (5)$$

Where, QCAL is the quantized calibrated pixel value in DN, Grescale is band-specific rescaling gain factor in (watts/(meter squared*ster*μm))/DN, Brescale is band-specific rescaling bias factor in watts/(meter squared*ster*μm), LMIN is the spectral radiance that is scaled to QCALMIN in watts/(meter squared*ster*μm), LMAX is the spectral radiance that is scaled to QCALMAX in watts/(meter squared*ster*μm), QCALMIN is the minimum quantized calibrated pixel value (corresponding to LMIN) in DN and QCALMAX is the maximum quantized calibrated pixel value (corresponding to LMAX) in DN.

Table1 summarize the TM spectral range, post-calibration dynamic ranges (LMIN and LMAX scaling parameters, the corresponding rescaling gain (Grescale) and rescaling bias (Brescale) values). QCALMIN=0 for data processed using NLAPS and QCALMIN=1 for data processed using LPGS.

Table 1

### 2.2.3 Conversion spectral radiance to brightness temperature ($L_\lambda$-to-T):

Thermal band data (band 6 on TM) can be converted from spectral radiance to effective brightness temperature. The brightness temperature assumes that the Earth's surface is a black body (spectral emissivity of the black body is 1).Thermal radiance values were converted from spectral radiance to brightness temperature using the thermal constants by following equation:

$$T = \frac{K_2}{\ln\left(\frac{K_1}{L_\lambda}+1\right)} \qquad (6)$$

Where,$T$ = Satellite brightness temperature (Kelvin), $L_\lambda$= TOA spectral radiance, $K_1$= Calibration constant 1 from the metadata, $K_2$= Calibration constant 2 from the metadata (Table 2).

Table 2

### 2.2.4 Conversion brightness temperature to land surface temperature (T-to-LST)

Since brightness temperature (T) is a blackbody temperature, the final step is the spectral emissivity according to the nature of the surface by temperature correction (Weng et. al. 2004):

$$ST = \frac{T_B}{1+\left(\lambda*{T_B}/\rho\right)*Ln\varepsilon} \qquad (7)$$

Where,$T_B$= Satellite rightness temperature (Kelvin), $\lambda$= Wavelength of emitted radiance (11.5 $\mu m$), $\varepsilon$= Land surface emissivity,$\rho = h*c/\sigma$= 1.438*$10^{-2}$mK ($\sigma$= Boltzmann constant= 1.38*$10^{-23}$ J/K, h= Planck's constant= 6.626*$10^{-34}$Js, c= velocity of light= 2.998*$10^8$ m/s).

Finally, derived land surface temperature in Kelvin was converted to Celsius by subtracting from 273.15.



For processing satellite data and database building, Gauss-Krueger coordinate system was selected. Following land classification was accepted using the Landsat TM data: asphalt road, park and green spaces, waste land and bare soil, fallow land, residential (urban), commercial and industrial (Javed Mallick et al., 2008). Classification was performed on Landsat TM for spectral separability of the land use classes existing in the study area. Finally, the relationship between land use classes, surface temperature and NDVI in Yazd city was analyzed in detail.

The False Colour Composite imagery of Landsat TM data (08th Aug, 1998 and 06th Aug, 2009) covered the study area was produced in ArcGIS environment (Figure 2). Land use classes were selected in the False Colour Composite imagery for supervised classification of image (Figure 3). Different land use types were classified using Maximum Likelihood classification.

### 2.2.5 Statistical methods:

The error matrix is used as an analytical statistical technique. The simplest descriptive statistical indicator is overall accuracy which can be calculated using error matrix. Producer's accuracy is the probability that a pixel in the classified image is placed in the same class on land. In the calculation, correct pixels total in each class is divided by the pixels total of that class as derived from the reference data. The producer's accuracy indicates the probability of a reference pixel being correctly classified. In the user's accuracy the correct pixels total in a land use class is divided by the total number of pixels that were classified in that class. The user's accuracy is indicative of the probability that a pixel classified on the image actually represents that category on the ground (Story and Congalton, 1986). User's accuracy is the probability that a specific class of land was classified in the same class on the classified image (Figuers 4, 5 and Tables 4, 5).

Validation of the obtained temperatures from 08th Aug, 1998 and 06th Aug, 2009 images was performed using the Root-Mean-Square Error (RMSE) and Coefficient of Determination ($R^2$) between the measured (ground-based) and predicted (by satellite data) land surface temperatures (Xiaolei Yu, 2014) (Figure 8 and Table 7).

### 3. Results and discussions

### 3.1 Analysis of Land use:

By classifying land uses of study area, six land-use types for study area were considered: asphalt roads, parks and green spaces, waste land and bare soil, fallow land, residential, commercial and industrial areas. The land use distribution in Yazd city is described in table 3. The classified images show that the agricultural lands were classified as fallow land because of similar values of their spectral reflectance. In the classified images it also was observed that in some places of study area fallow lands have been combined with waste land and bare soil. By comparing area percentage values of different land use classes can be concluded that land use types of study area were significantly converted in the 11-year period (Table 3). Table 3 shows that the asphalt roads, commercial, industrial and residential (Urban) classes of land use were increased, that's mean these land use categories have more area in 2009 compared to the year 1998. However, parks, green spaces, fallow lands, waste lands and bare soils were decreased during the time. The study results clearly show that residential areas had a most changes compared to the other classes of land use. The main changes include conversion parks, green spaces, bare soils and fallow lands to residential areas.

Figure 2

Figure 3

Table 3



Table 4

Figure 4

For example, in the 08th Aug, 1998 image by comparing the reference and classified data in the able 4, it was observed that although 100% of the asphalt roads were being correctly identified as asphalt roads, only 90.9% of the areas called asphalt roads are actually

asphalt roads. And in the 06th Aug, 2009 image by comparing the reference and classified data, it was observed that 95.24% of the asphalt roads were being correctly identified as asphalt roads, while 86.96% of the areas called asphalt roads are actually asphalt roads (Table 5). Water class does not exist in the land use classification because of the lack of surface water resources in the study area. The Kappa coefficient values of the 08th Aug, 1998 and 06th Aug, 2009 classified images were 0.964 and 0.946, respectively. The error matrixes indicate an overall accuracy of 0.965 and 0.948 for 08th Aug, 1998 and 06th Aug, 2009 classified images,

respectively. These values obtained from analysis of the referenced data (ground-based data) and the classification output (Tables 4 and 5). Based on the results, the classified images have significantly good accuracy over different land use types (Figures 4 and 5).

Table 5

Figure 5

### 3.2 Analysis of vegetation situation and NDVI index

The spatial distribution of NDVI values from the Landsat TM image can be seen in figure 6.The 08th Aug, 1998 NDVI values are in the range of -0.2 to 0.59, having a mean value of 0.195, and the 08th Aug, 1998 NDVI values are in the range of -0.17 to 0.41, having a mean value of 0.12 (Figure 6).

Figure 6

In the figure it is shown that low values of NDVI (light green area) correspond to waste land, bare soil, commercial, industrial and residential areas on the different parts mainly in the southern and western parts of the study area. High values of NDVI (dark green) that were observed in the central, north and southwest parts of the images correspond to parks and green spaces. The medium NDVI values were observed over fallow land and asphalt roads, in the central, north and southwest parts of the study area. By comparing NDVI of two different times (1998 and 2009) we concluded that NDVI values decreased over the studied period of

time because of plant-covered surfaces reduction. The maximum values of derived emissivity are observed over parks and green spaces. The emissivity values of parks and green spaces of the year 1998 image are from 0.987 to 0.99. In addition, emissivity values of the fallow land class are found in the range of 0.986 to 0.987. Asphalt roads, commercial and industrial, residential, waste land and bare soil have very similar emissivity values from 0.986 to 0.9863. The emissivity values of parks and green spaces in the year 2009 image are from 0.982 to 0.99, emissivity values of the fallow land class are from 0.979 to 0.982 and emissivity

values of the asphalt roads, commercial and industrial, residential, waste land and bare soil are from 0.97 to 0.979.Wheat and barley is mainly grown in the agricultural lands of the study area (seasonal plants) that in the performed classification are placed on the fallow land class, however, these agricultural products were harvested in May month.

### 3.3 Land surface temperature analysis

In the present study, land surface temperature was retrieved by Temperature Emissivity Separation (TES) algorithm from TIRS

data of the Landsat Thematic Mapper (TM). The spatial distribution of surface temperature of the years 1998 and 2009 images is shown in the figure 7. Surface temperature of the year 1998 LST image ranged from 27.1 to 45.57°C (mean of 36.34°C), and





surface temperature of the year 2009 LST image ranged from 30.01 to 45.57°C (mean of 37.79°C). In the figure 7 high surface temperatures are shown by the dark red areas, this means that outlying parts of the city have a temperature higher than central part. This can be due to many cooling devices which are used in residential areas and also due to keeping parks and green spaces in the central part and the destruction of vegetation in the outlying parts of the city. The results indicated that average temperature of

5 Yazd city increased from 36.34 to 37.79°C. The temperature increasing has different reasons, including vegetation loss and land use change in the area. Developments in asphalt roads, residential and commercial areas increased dramatically between 1998 and 2009, and vegetation covers has been reduced. The combination of mentioned factors caused an increase the overall temperature of the city.

Figure 7

In the year 1998 classified image, bare soil and waste land (mean value 38.61°C), residential (mean value 38.22°C) classes have maximum values of surface temperature, and parks and green spaces have minimum values (mean value 34.47°C).In the year 2009, bare soli and waste land (mean value 39.06°C), commercial and industrial (mean value 39.83°C) classes have maximum values of surface temperature, and parks and green spaces have minimum values (mean value 36.69°C). It is generally observed that surface temperature has been increased in the all types of land use, but the greatest increase is registered in the commercial and industrial

sites (Figure 7 and Table 6). It should be noted that residential areas had a slight decrease in the surface temperature. This is probably due to many cooling devices which are used in residential areas of the Yazd city, especially in the summer.

Table 6

For the investigated area, results show that the land surface temperatures retrieved from TES algorithm using Landsat TM sensor data have the high accuracy with RMSE 0.902°C and 0.866°C for 08th Aug, 1998 and 06th Aug, 2009, respectively. The Coefficient

of Determination ($R^2$) between the measured and predicted temperatures is 0.983 and 0.991 for 08th Aug, 1998 and 06th Aug, 2009, respectively (Figure 8). The high coefficient of determination ($R^2$) and low RMSE indicate that the study results have high accuracy.

Figure 8

Table 7

**3.4 Statistical analysis of the relationship between land surface temperature, normalized difference vegetation index (NDVI) and land use changes**

In the present study, strong relation was observed between NDVI index, surface temperature and land use types changes (Figure 9 and Table 8, 9). The spatial variations of surface temperature are affected by the conversion of land for human-dominated use (land use change) and vegetation cover. Values of NDVI index have an inverse relation with land surface temperature. This means that

decrease in NDVI corresponded to an increase in temperature of land use types and vice versa. The strongest inverse relationship between surface temperature and NDVI values was observed in the fallow lands, parks and green spaces. By reducing vegetation density (in this study waste land and bare soil), inverse relation between surface temperature and NDVI value was weaker. By increasing vegetation cover, NDVI index values increased and surface temperature decreased. Also, an increase in temperature and a decrease in NDVI values were observed by increasing waste and barren lands in area. There is a linear regression between

surface temperature and NDVI (Figure 9); therefore, surface temperature can be estimated if NDVI values are known with reasonable accuracy. According to the received results, NDVI values were decreased during the study time (Table 8).



Figure 9

Table 8

Land use types changes were causing surface temperature increasing, and temperature increasing subsequently was causing changes of NDVI and vegetation covers in the study area. The significant differences less than 0.01 between NDVI values and temperatures of land use types for 08th Aug, 1998 and 06th Aug, 2009, cases were obtained by statistical analysis of NDVI values and surface temperatures. The significant differences less than 0.01 indicate that with the probability of 99 % temperature increasing (was caused by Land use changes) were causing a decrease in the NDVI values (Table 9).

Table 9

The decrease in vegetation cover and an increase in residential areas are the main reasons of decreasing NDVI values of the Yazd city. Land use gradual changes during the time are causing the temperature change. In the Yazd city surface temperature has increased in the result of increases in asphalt roads, commercial, industrial and residential areas and a decrease in parks, green spaces and fallow land classes in 2009 compared to the year 1998. However, waste land and bare soil decreased, but mutually asphalt roads, commercial, industrial and residential areas increased, and these changes caused a rise in temperature. Other studies also confirm that the LST and NDVI changes are due to changes of the vegetation cover and residential areas (Gong Z. et al., 2015; Sandra E. et al., 2015; Valor E. et al., 1996; Wei L. et al., 2015 and Javed Mallick et al., 2008).

**4 Conclusions**

The study results show that LST, NDVI and surface emissivity can be estimated using Landsat TM sensor imagery with high accuracy. Calculate surface temperature and NDVI is important in the earth studies including global environmental change, urban climate change and urbanization. Different land use types of urban areas can be studied by estimating NDVI index and land surface temperature values. This paper explored the spatial and temporal relationship between NDVI, LST and land use types. It was found, that in the Yazd city combination of vegetation cover decreasing, residential areas increasing and other changes in land use was directly causing surface temperature increasing. By comparing two different times (1998 and 2009), we concluded that the average surface temperature of the Yazd city has risen 1.45 degrees Celsius. Considering the impacts of land use types changes and vegetation cover decreasing on the rising surface temperature, the role of human activities becomes more and more evident in climate change. According to the results, simultaneous analysis of the NDVI, LST and land use types changes is ideal for the study of urban environment and climate change, because of dealing directly with vegetation cover and surface temperature. Based on the study results, the highest percentage of Yazd area is residential areas, fallow lands, waste land and bare soil, and this is directly related to climate situation (arid and semi-arid climate) and human activities.

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





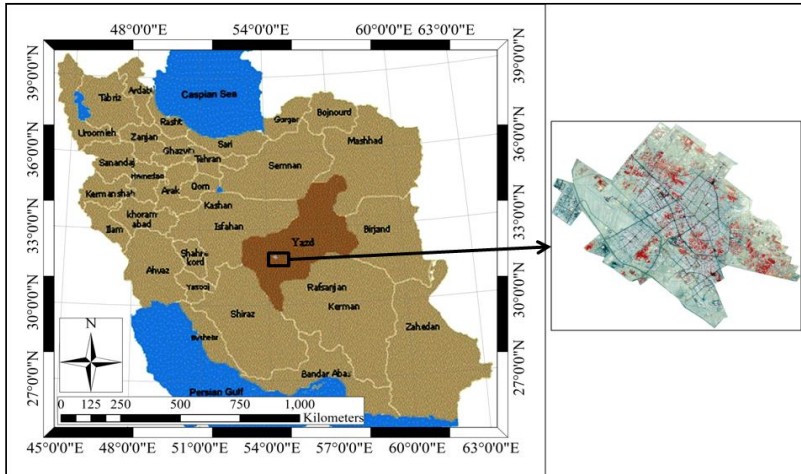

Figure 1. Yazd city location in Iran

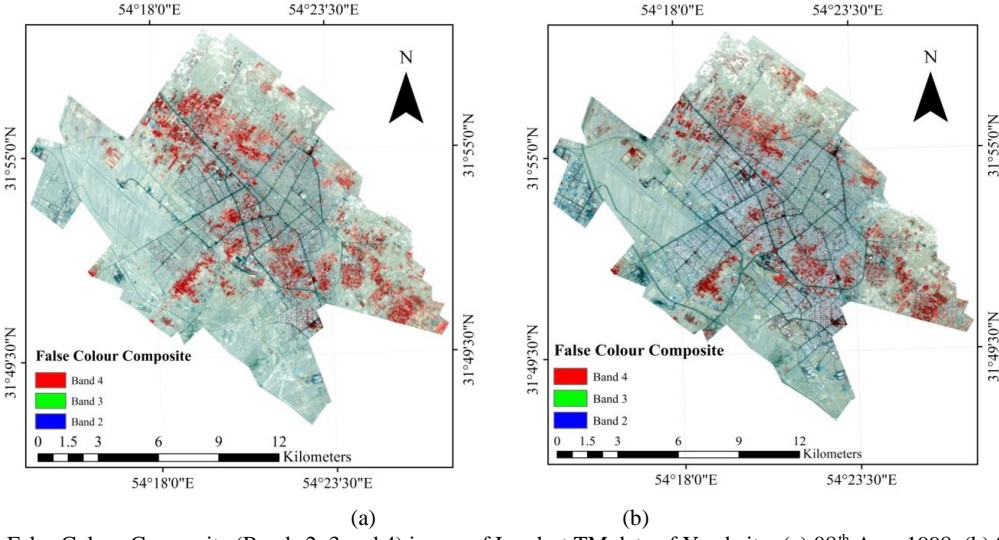

| (a) | (b) |

5      Figure 2. False Colour Composite (Bands 2, 3 and 4) image of Landsat TM data of Yazd city: (a) 08th Aug, 1998, (b) 06th Aug, 2009.




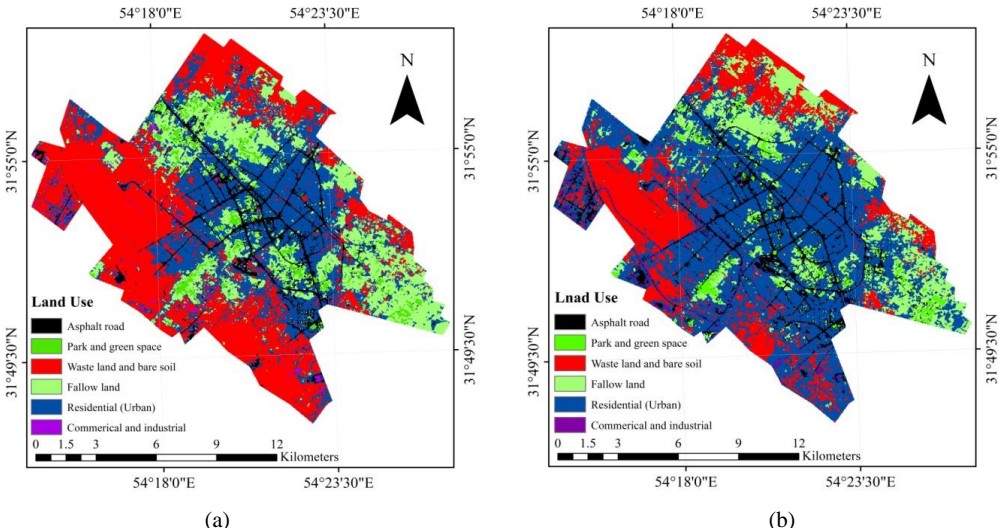

(a)                                                              (b)

Figure 3. Land use classified image of Yazd city by Maximum Likelihood classification: (a) 08th Aug, 1998, (b) 06th Aug, 2009.

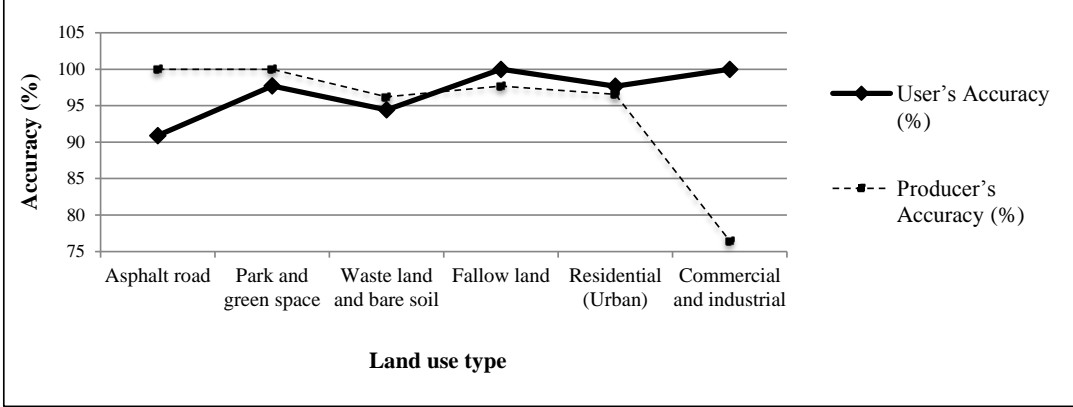

Figure 4. Land use classes User's and Producer's Accuracy of the 08th Aug, 1998 Landsat TM image.

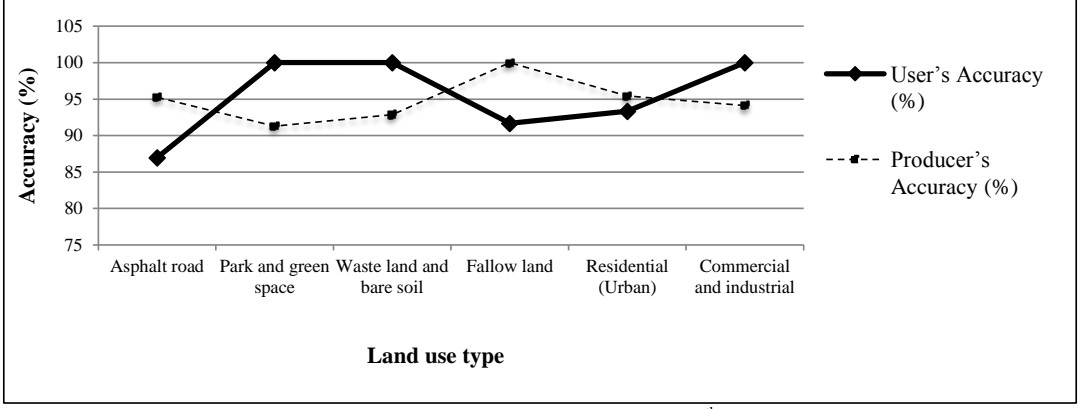

Figure 5. Land use classes User's and Producer's Accuracy of the 06th Aug, 2009 Landsat TM image.



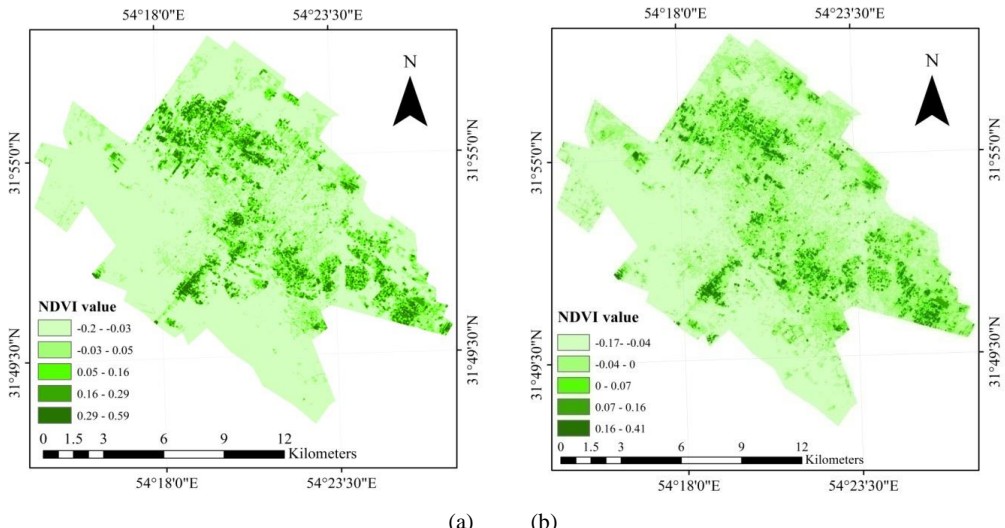

(a)          (b)

Figure 6. Spatial distribution of NDVI index obtained from Landsat TM data: (a) 08th Aug, 1998, (b) 06th Aug, 2009.

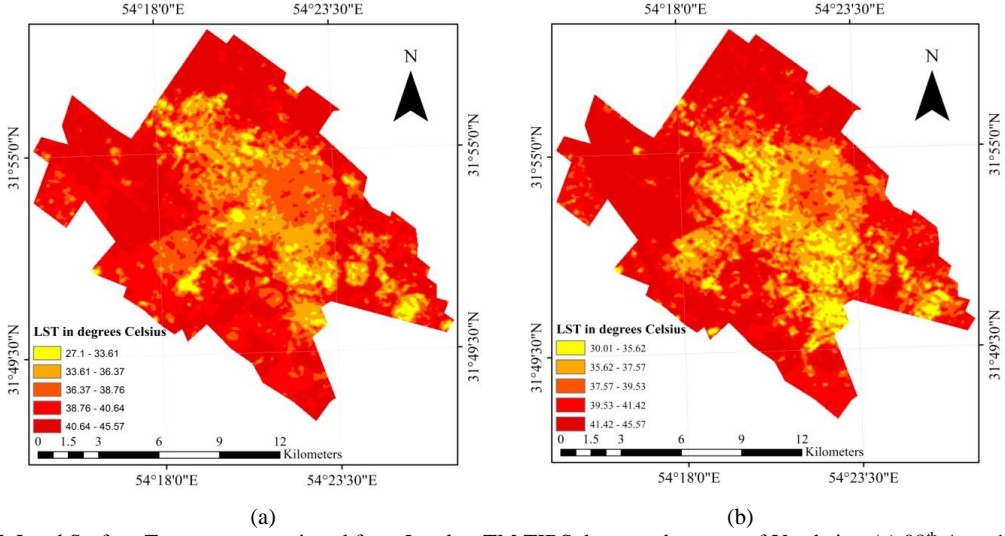

5                                  (a)                                                            (b)

Figure 7. Land Surface Temperature retrieved from Landsat TM TIRS data at urban area of Yazd city: (a) 08th Aug, 1998, (b) 06th Aug, 2009.





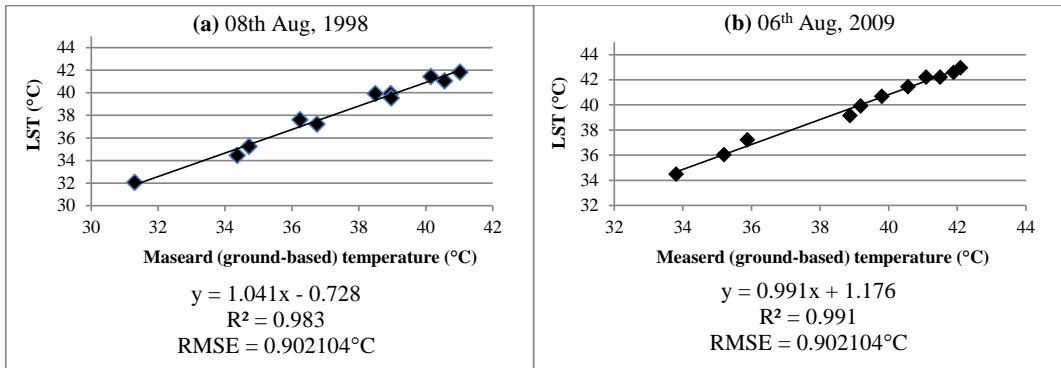

Figure 8. Linear relationship between ground-based measured temperature and land surface temperature (LST) of Landsat TM sensor: (a) 08th Aug, 1998, (b) 06th Aug, 2009.

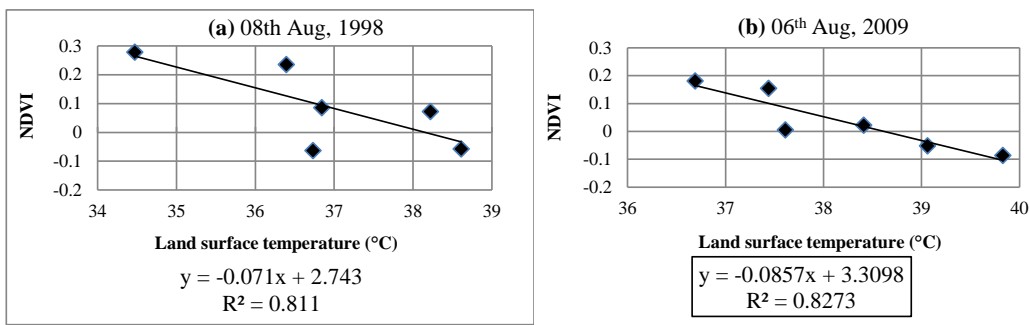

Figure 9. Linear relationship between NDVI values and land surface temperature (LST) of Landsat TM sensor: (a) 08th Aug, 1998, (b) 06th Aug, 2009.

Table 1. TM spectral range and post-calibration dynamic ranges.

| Band number | Spectral range (μm) | Center wavelength | LMIN (w/(m²*sr*μm)) | LMAX (w/(m²*sr*μm)) | Grescale ((w/(m²*sr*μm))/DN) | Brescale (w/(m²*sr*μm)) |
|---|---|---|---|---|---|---|
| 1 | 0.452-0.518 | 0.485 | -1.52 | 169 | 0.671339 | -2.19 |
|  |  |  | -1.52 | 193 | 0.765827 | -2.29 |
| 2 | 0.528-0.609 | 0.569 | -2.84 | 333 | 1.322205 | -4.16 |
|  |  |  | -2.84 | 365 | 1.448189 | -4.29 |
| 3 | 0.626-0.693 | 0.660 | -1.17 | 264 | 1.043976 | -2.21 |
| 4 | 0.776-0.904 | 0.840 | -1.51 | 221 | 0.876024 | -2.39 |
| 5 | 1.567-1.784 | 1.676 | -0.37 | 30.2 | 0.120354 | -0.49 |
| 6 | 10.45-12.42 | 11.435 | 1.2378 | 15.3032 | 0.055376 | 1.18 |
| 7 | 2.097-2.349 | 2.223 | -0.15 | 16.5 | 0.065551 | -0.22 |

Table 2. TM thermal band calibration constants.

|  | Constant 1 - K1 (watts/(meter squared*ster*μm) | Constant 2 - K2 (Kelvin) |
|---|---|---|
| Landsat 5 | 607.76 | 1260.56 |

Table 3. Land use distribution of Yazd city using Maximum Likelihood classification.

| Land use | Image of the 08th Aug, 1998 | | Image of the 06th Aug, 2009 | | Difference between images of 1998 and 2009 | |
|---|---|---|---|---|---|---|
|  | Study area (km²) | Study area (%) | Study area (km²) | Study area (%) | km² | % |
| Asphalt roads | 13.586029 | 6.9 | 18.358887 | 9.33 | 4.772858 | 2.43 |
| Parks and green spaces | 6.558943 | 3.33 | 4.935898 | 2.51 | -1.623045 | -0.82 |





| | | | | | |
|---|---|---|---|---|---|
| Waste land and bare soil | 77.454838 | 39.34 | 42.078257 | 21.37 | -35.376581 | -17.97 |
| Fallow land | 38.03299 | 19.32 | 28.8658 | 14.66 | -9.16719 | -4.66 |
| Residential (Urban) | 59.359028 | 30.15 | 99.858542 | 50.72 | 40.499514 | 20.57 |
| Commercial and industrial | 1.896892 | 0.96 | 2.779129 | 1.41 | 0.882237 | 0.45 |

Table 4. Error matrix used to assess the accuracy of a classification of the 08th Aug, 1998 Landsat TM image.

| | Land use types | Asphalt road | Park and green space | Waste land and bare soil | Fallow land | Residential (Urban) | Commercial and industrial | Total | User's Accuracy (%) |
|---|---|---|---|---|---|---|---|---|---|
| | | | | | Reference data | | | | |
| Classified data | Asphalt road | 40 | 0 | 0 | 0 | 0 | 4 | 44 | 90.9 |
| | Park and green space | 0 | 43 | 0 | 1 | 0 | 0 | 44 | 97.72 |
| | Waste land and bare soil | 0 | 0 | 51 | 0 | 3 | 0 | 54 | 94.44 |
| | Fallow land | 0 | 0 | 0 | 42 | 0 | 0 | 42 | 100 |
| | Residential (Urban) | 0 | 0 | 2 | 0 | 84 | 0 | 86 | 97.67 |
| | Commercial and industrial | 0 | 0 | 0 | 0 | 0 | 13 | 13 | 100 |
| | Total | 40 | 43 | 53 | 43 | 87 | 17 | 283 | - |
| | Producer's Accuracy (%) | 100 | 100 | 96.23 | 97.67 | 96.55 | 76.47 | - | - |

Table 5. Error matrix used to assess the accuracy of a classification of the 06th Aug, 2009 Landsat TM image.

| | Land use types | Asphalt road | Park and green space | Waste land and bare soil | Fallow land | Residential (Urban) | Commercial and industrial | Total | User's Accuracy (%) |
|---|---|---|---|---|---|---|---|---|---|
| | | | | | Reference data | | | | |
| Classified data | Asphalt road | 20 | 0 | 0 | 0 | 2 | 1 | 23 | 86.96 |
| | Park and green space | 0 | 21 | 0 | 0 | 0 | 0 | 21 | 100 |
| | Waste land and bare soil | 0 | 0 | 26 | 0 | 0 | 0 | 26 | 100 |
| | Fallow land | 0 | 2 | 0 | 22 | 0 | 0 | 24 | 91.67 |
| | Residential (Urban) | 1 | 0 | 2 | 0 | 42 | 0 | 45 | 93.33 |
| | Commercial and industrial | 0 | 0 | 0 | 0 | 0 | 16 | 16 | 100 |
| | Total | 21 | 23 | 28 | 22 | 44 | 17 | 155 | - |
| | Producer's Accuracy (%) | 95.24 | 91.3 | 92.86 | 100 | 95.45 | 94.12 | - | - |

Table 6. Land surface temperature of different land use categories of Yazd city.

| Land use and land cover | Min. temperature (°C) | | Max. temperature (°C) | | Mean (°C) | |
|---|---|---|---|---|---|---|
| | 08th Aug, 1998 | 06th Aug, 2009 | 08th Aug, 1998 | 06th Aug, 2009 | 08th Aug, 1998 | 06th Aug, 2009 |
| Asphalt roads | 29.6 | 31.24 | 44.08 | 45.57 | 36.84 | 38.41 |
| Parks and green spaces | 27.1 | 30.42 | 41.83 | 42.96 | 34.47 | 36.69 |
| Waste land and bare soil | 31.65 | 33.66 | 45.57 | 44.46 | 38.61 | 39.06 |
| Fallow land | 27.94 | 30.42 | 44.83 | 44.46 | 36.39 | 37.44 |
| Residential (Urban) | 31.24 | 30.01 | 45.2 | 45.2 | 38.22 | 37.61 |
| Commercial and industrial | 31.24 | 34.46 | 42.21 | 45.2 | 36.73 | 39.83 |

Table 7. Accuracy assessment of land surface temperature derived from Landsat TM sensor data using calculation RMSE and $R^2$.

| LST Landsat TM sensor images | root-mean-square error (RMSE) | Coefficient of Determination ($R^2$) |
|---|---|---|
| 08th Aug, 1998 | 0.902 | 0.983 |





| 06th Aug, 2009 | 0.866 | | 0.991 | |
|---|---|---|---|---|

Table 8. Relation between surface temperatures with NDVI of the different land use categories.

| Land use and land cover | Mean temperature (°C) | | NDVI value (mean) | |
|---|---|---|---|---|
| | 08th Aug, 1998 | 06thAug, 2009 | 08th Aug, 1998 | 06th Aug, 2009 |
| Asphalt roads | 36.84 | 38.41 | 0.085 | 0.023 |
| Parks and green spaces | 34.47 | 36.69 | 0.278 | 0.182 |
| Waste land and bare soil | 38.61 | 39.06 | -0.058 | -0.051 |
| Fallow land | 36.39 | 37.44 | 0.235 | 0.155 |
| Residential (Urban) | 38.22 | 37.61 | 0.072 | 0.006 |
| Commercial and industrial | 36.73 | 39.83 | -0.063 | -0.086 |

Table 9. Variance analysis of relationship between land surface temperature and normalized difference vegetation index (NDVI) retrieved from Landsat TM data.

| Landsat TM data date | Source | Sum of Squares | df | Mean Square | Significant difference |
|---|---|---|---|---|---|
| 08th Aug, 1998 | Between Groups | 0.510 | 5 | 0.102 | 0.000 |
| | Within Groups | 0.000 | 24 | 0.000 | |
| | Total | 0.510 | 29 | | |
| 06thAug, 2009 | Between Groups | 0.295 | 5 | 0.059 | 0.000 |
| | Within Groups | 0.000 | 24 | 0.000 | |
| | Total | 0.295 | 29 | | |