# Peer review of "Derivation of land surface temperature from Landsat Thematic Mapper (TM) sensor data and analyzing relation between land use changes and surface temperature"

_Solid Earth, 2016_

## Referee Comment (RC1) · Dr. Miller (Referee) · 15 Mar 2016

General Comments

The present manuscript tells an interesting story of land use change, particularly the loss of vegetation as detected by NDVI, corresponding with increases in land surface temperature, all detectable by remote sensing. It is novel in its combination of methods and application to a hot and semi-arid to arid region. However, there are several grammatical and organizational issues that should be addressed before acceptance for publication. One critique in this regard is that the writing should expand on what is

unique to this study and direct readers to the references for more information on the established methods upon which this study is building upon. Second, the organization needs to more clearly separate the tests validating the spatial models and the analysis for how land use has changed and its implications for local surface temperature. However, the main issue preventing acceptance is the lack of information about the data collected on the ground, used to calibrate and validate the models. Without this, the basis for the conclusions cannot be evaluated.

Specific Comments

1. The title could be improved. Perhaps something like "Using Landsat to detect change in land surface temperature in relation to land use change in Yazd, Iran" may more clearly communicate the focus of the research.

2. The abstract needs to be more to the point. The background in the first half can go in the paper's introduction, but the abstract needs to focus on the work done in this research and its outcomes.

3. The introduction is currently one long paragraph. This should be broken into multiple paragraphs.

4. The majority of the methods text is spent on reviewing equations presented in previous papers, upon which the present research is building upon. This should be condensed and more heavily cited, allowing the reader to refer to those papers for more information.

5. After reducing the content in the methods section about previously established equations, text needs to be added fully explaining the original work in this study. How was information for the calibration and validation points collected? How many were there in each category and where were they located? Without this information it is difficult to assess if the results are sufficient to support the interpretations and conclusions.

6. The results and discussion section is poorly organized, mixing model validation
and interpretations in a confusing way. I would recommend making the results and discussion separate. This way, the results can focus on the validation of the models converting the Landsat data into information about land use change, vegetation vigor (NDVI), and land surface temperature. The discussion can then go into interpretations of the relationships between these things.

7. Be consistent in the use of decimal places. Significant digits may not be clear in these circumstances of modeling, but thousandths place is likely not meaningful.

8. Figures 4 and 5 are not illustrating sequential relationships between the attributes of the x-axis. Therefore, a line graph is not appropriate. Recommend using a grouped bar graph.

Technical Corrections

P1, L20 and 21 – remove "the" before singular place names, i.e. "Iran" and "Yazd." Also, one can use "city of Yazd" instead of "Yazd city" to clarify that Yazd is a city, but thereafter, only "Yazd" is needed. If referring to the province of Yazd, then "Yazd Province" would be used.

P1, L24 – replace "relation" with "relationships"

P1, L28 – insert "land use" after "industrial"

P1, L33 – "The main problem" is a strong statement, "A problem" may be more appropriate, unless there really are no other big problems for the city of Yazd

P1, L36 – delete "that"

P1, L37 – replace "during" with "for cooling" and delete "cooling" after "buildings"

P2, L1 – replace "mapping" with "map"

P2, L2 – Insert "The" before "LST indicator"

P2, L11 – move "different" to before "land surface"

P2, L13 – change "LST index provide" to "The LST index provides"

P2, L36 – the I in NDVI stands for index, so delete "index" after "NDVI"

P3, L7 – insert "The" before "study area"

P3, L9 – change "amount" to "amounts" and "precipitations" to "precipitation"

P3, L11 – insert "the" before "Caspian Sea" (I realize this appears to contradict the earlier correction for city and country names, but that's the way English is. Here are some helpful guides: https://www.englishclub.com/grammar/nouns-proper-no-the.htm https://www.englishclub.com/grammar/nouns-proper-the.htm

P3, L15 – insert "the" before "present study"

P3, L17 – delete "pixel scale (pixel" and the following ")", to read "Natural surfaces at the resolution of 30 meters are heterogeneous"

P3, L23 – insert "the" before "following"

P3, L35 and L36 – delete "The" before "high" and replace "vegetation density and health" with "dense and healthy vegetation"

P4, L1 – insert "of" before "calibrated"

P5, L27 – insert "between images, it" after "land use classes"

P5, L29 and L30 – delete "that's mean these land use categories have more area in 2009 compared to the year 1998" (grammar would need fixed, but is redundant from the first half of the sentence anyway)

P5, L31 – delete "clearly" (subjective), replace "had a" with "increased", and delete "changes"

P5, L32 – insert "of" before "parks"

P6, L1 – non sequitur. The following content is an evaluation of the classification

model's accuracy, not an example of the previous paragraph's evaluation of land use change.

P6, L1 – Once the exact date of the image has been defined in the methods section, there is no need to repeat the full date. Referring to the respective images as the "1998 image" and "2009 image" will suffice and will be more concise.

Figure 8 - x-axis labels have misspellings. change to "measured"

Please use these specific grammar tips to improve English writing throughout the paper.

---

## Author Comment (AC1) · 25 Mar 2016

**Using Landsat Thematic Mapper (TM) sensor to detect change in land surface temperature in relation to land use change in Yazd, Iran**

S. Zareie[1], H. Khosravi[*2], A. Nasiri[3]

[1] Ph.D. student of GIS & Remote Sensing, Institute of Earth Sciences, Saint Petersburg State University, Saint Petersburg, Russian federation

[2] Department of Arid and Mountainous Regions Reclamation, Faculty of Natural Resources, College of Agriculture & Natural Resources, University of Tehran, Karaj, Iran (hakhosravi@ut.ac.ir)

[3] Department of Ecology and Environmental Management, Protection of the Natural Resource and Environment, Land Cadastre Faculty, State University of Land Use Planning, Moscow, Russia

*Correspondence to*: h. khosravi (hakhosravi@ut.ac.ir)

*Abstract*

Land surface temperature (LST) is one of the key parameters in the physics of land surface processes from local to global scales, and it is one of the indicators of environmental quality. Evaluation of the surface temperature distribution and its relation with existing land use types are very important to investigate the urban microclimate. In the arid and semi-arid regions, understanding the role of land use changes in the formation of urban heat islands is necessary to provide urban planning to control or reduce surface temperature. Internal factors and environmental conditions of Yazd city have important role in the formation of special thermal conditions in Iran. In this paper, we used the Temperature Emissivity Separation (TES) algorithm for LST retrieving from TIRS data of the Landsat Thematic Mapper (TM). The Root Mean-Square Error (RMSE) and Coefficient of Determination ($R^2$) were used for validation of retrieved LST values. The RMSE of 0.9°C and 0.87°C and $R^2$ of 0.98 and 0.99 were obtained for 1998 and 2009 images, respectively. Land use types of the city of Yazd were identified and relationships between land use types, land surface temperature and NDVI index were analyzed. The Kappa coefficient and overall accuracy were calculated for accuracy assessment of the land use classification. The Kappa coefficient values are 0.96 and 0.95 and the overall accuracy values are 0.97 and 0.95 for 1998 and 2009 classified images, respectively. The results of this study showed that optical and thermal remote sensing methodologies can be used to research urban environmental parameters. Finally, it was found that special thermal conditions in Yazd were formed by land use changes. Increasing the area of asphalt roads, residential, commercial and industrial land use types and decreasing the area of the parks, green spaces and fallow lands types of land uses in Yazd caused a rise in surface temperature during the 11-year period. The results showed an increase of 1.45 degrees Celsius of the average surface temperature.

**Keywords:** Yazd, NDVI, Landsat Thematic Mapper, LST, Land use.

**1 Introduction**

A problem in the urban areas is surface temperature increasing due to conversion of vegetated surfaces into asphalt roads, residential, commercial and industrial areas. Nowadays, climate change in the cities is occurring by anthropogenic activities and land use changes. The atmospheric conditions of the urban areas, land surface temperature, warming, evaporation and absorption

of solar radiation may be changed by anthropogenic changes. The study of surface temperature in the cities located in the arid and semi-arid areas is necessary, because high temperature leads to energy consumption for cooling buildings, which is economically very costly especially in the warm months of year.

[revised manuscript text omitted]

**2.3 Calibration and validation of Landsat TM LST and land use classification**

The validation of Land Surface Temperature (LST) of satellite products is performed by using ground-based measurements of surface temperature. In the present study, ground-based data of land surface temperature received from Yazd Meteorological Bureau were used to validate LST of Landsat TM sensor. Measuring land surface temperature was performed using thermometers with SMT160 temperature sensor. The thermometers have a temperature range of -45 to 130 ° C. The most important characteristics of the SMT160 temperature sensor include absolute accuracy ∓0.7 ° C and measurement range 175 ° C. Geographical characteristics of the measurement points can be seen in Table 3.

Table 3

[revised manuscript text omitted]

**Discussions**

This paper proposed the TES algorithm to obtain LST from Landsat TM and change detection in land surface temperature in relation to land use change. The spatial scales of TIRS pixel (100 m), land use and NDVI products (30 m) can reduce accuracy of the results. The classified images show that the agricultural lands were classified as fallow land, and it also was observed that in some places of study area fallow lands have been combined with waste land and bare soil because of similar values of their spectral reflectance.  As a result, the main factor of the conversion of land use types of study area in the 11-year period is the human activities for urban growth. Meanwhile, water class does not exist in the land use classification because of the lack of surface water resources in the study area. Based on the results, the classified images have significantly good accuracy over different land use types (Figures 4 and 5). As mentioned in Section 3.2, the NDVI values decreased over the studied period of time (1998-2009) because of plant-covered surfaces reduction. Wheat and barley is mainly grown in the agricultural lands of the study area (seasonal plants) that in the performed classification are placed on the fallow land class because of similar values of spectral reflectance of these products and fallow lands. It should be noted that 
[revised manuscript text omitted]

| Measurement point | Latitude | Longitude | 08th Aug, 1998 Ground-based measured temperature(° C) | 08th Aug, 1998 Landsat TM LST | 06th Aug, 2009 Ground-based measured temperature(° C) | 06th Aug, 2009 Landsat TM LST |
|---|---|---|---|---|---|---|
| 1(Yazd synoptic station) | 31°53'59.9994"N | 54°16'59.9982"E | 31.1 | 32.06 | 33.8 | 34.5 |
| 2 | 31°53'51.792"N | 54°14'20.833"E | 38.95 | 39.93 | 40.56 | 41.45 |
| 3 | 31°53'42.415"N | 54°15'52.937"E | 38.97 | 39.55 | 41.09 | 42.21 |
| 4 | 31°54'7.576"N | 54°15'53.826"E | 40.56 | 41.07 | 41.89 | 42.59 |
| 5 | 31°54'53.781"N | 54°15'55.217"E | 40.15 | 41.45 | 41.5 | 42.21 |
| 6 | 31°54'51.239"N | 54°17'33.628"E | 41.02 | 41.83 | 42.09 | 42.96 |
| 7 | 31°54'48.834"N | 54°19'5.99"E | 36.75 | 37.22 | 39.8 | 40.69 |
| 8 | 31°55'49.672"N | 54°19'8.177"E | 36.24 | 37.61 | 39.18 | 39.93 |
| 9 | 31°54'46.463"N | 54°20'44.16"E | 34.72 | 35.26 | 35.2 | 36.05 |
| 10 | 31°53'40.635"N | 54°23'49.896"E | 38.49 | 39.93 | 38.87 | 39.16 |
| 11 | 31°52'50.098"N | 54°22'15.626"E | 34.37 | 34.46 | 35.88 | 37.22 |

Table 4. Land use distribution of Yazd city using Maximum Likelihood classification.

[revised manuscript text omitted]

---

## Referee Comment (RC2) · Anonymous Referee #2 · 3 May 2016

The paper used the Temperature Emissivity Separation (TES) algorithm for LST retrieving from TIRS data of the Landsat Thematic Mapper (TM). The results of this study found that special thermal conditions in Yazd city were formed by land use changes. Increasing the area of asphalt roads, residential, commercial and industrial types and decreasing the area of the parks, green spaces and fallow lands types of land uses in the Yazd city caused a rise in surface temperature during the 11-year period. However, the finding of this study would not be outstanding for the contribution to scientific progress.

[Figure]

The abstract should be concise to show the main finding and novelty of the paper. The structure of introduction is too chaotic. It should be reorganized to several paragraphs to introduce the relative studies and emphasize the objective of the study. It lacks a detailed description of the advantage and novelty of the study.

According to the result, it shows that the spatial variations of surface temperature are affected by the conversion of land for human-dominated use (land use change) and vegetation cover. However, why was the climate change not taken into consideration in the analysis? And what was the role of the climate change on the surface temperature?

It should be noted that residential areas had a slight decrease in the surface temperature. The manuscript explained it was probably due to many cooling devices which are used in residential areas of the Yazd city, especially in the summer. The explanation was not convincing.

Table 9: what is the "Within Groups"?

The paper needs to be thoroughly checked for grammar because the poor quality of the English in places makes the text hard to follow. Spelling and grammatical mistakes should be corrected. For example: Page2 L1: "mapping" revised to "map" Page3 L7: "km2" revised to the superscript of "2" Page7 L36: deleted " were" in "NDVI values were decreased"

---

## Author Comment (AC2) · 8 May 2016

Dear Reviewer in Chief,

My revised paper has been attached. It has been provided based on your comments. We tried to provide paper based on all of comments. The revised parts has been shown with different colors. If there is any problems do not hesitate to get in touch me.

What is the "Within Groups"? The formula for the one-way Variance Analysis of relationship between land surface temperature and normalized difference vegetation index (NDVI) F-test statistic is F=(explained variance)/(unexplained variance) or

[Figure]

F=(Between group variability)/(Within Group variability) "Within Group variability" is the "unexplained variance" that shows experimental error rate. $\sum\_in_i(Y\_i-Y)^2/((K-1))$ where denotes the sample mean in the $i$th group, $n_i$ is the number of observations in the $i$th group denotes the overall mean. $1$, $N-K$ degrees of freedom under the null hypothesis. The statistic will be large if the between-group variability is large relative to the within-group variability, which is unlikely to happen if the population means of the group. some measure of whether the variables within the group of variables are measuring the same thing. Correlation between groups so some measure, assuming that each group reflects one overall trait, of how each trait (group) is related to every other trait.

Best Regards,

Hassan Khosravi Assistant Professor Department of Arid and Mountainous Regions Reclamation, Faculty of Natural Resources, University of Tehran, Iran E-mail: hakhosravi@ut.ac.ir tel: +98 9128446358

Please also note the supplement to this comment: http://www.solid-earth-discuss.net/se-2016-22/se-2016-22-AC2-supplement.pdf

———————————————

[Figure]

**Supplement:**

**Using Landsat Thematic Mapper (TM) sensor to detect change in land surface temperature in relation to land use change in Yazd, Iran**

S. Zareie[1], H. Khosravi[*2], A. Nasiri[3]

[1] Ph.D. student of GIS & Remote Sensing, Institute of Earth Sciences, Saint Petersburg State University, Saint Petersburg, Russian federation
[2] Department of Arid and Mountainous Regions Reclamation, Faculty of Natural Resources, College of Agriculture & Natural Resources, University of Tehran, Karaj, Iran (hakhosravi@ut.ac.ir)
[3] Department of Ecology and Environmental Management, Protection of the Natural Resource and Environment, Land Cadastre Faculty, State University of Land Use Planning, Moscow, Russia

*Correspondence to*: h. khosravi (hakhosravi@ut.ac.ir)

**Abstract**

Land surface temperature (LST) is one of the key parameters in the physics of land surface processes from local to global scales, and it is one of the indicators of environmental quality. Evaluation of the surface temperature distribution and its relation with existing land use types are very important to investigate the urban microclimate. In the arid and semi-arid regions, understanding the role of land use changes in the formation of urban heat islands is necessary to provide urban planning to control or reduce surface temperature. Internal factors and environmental conditions of Yazd city have important role in the formation of special thermal conditions in Iran. In this paper, we used the Temperature Emissivity Separation (TES) algorithm for LST retrieving from TIRS data of the Landsat Thematic Mapper (TM). The Root Mean-Square Error (RMSE) and Coefficient of Determination ($R^2$) were used for validation of retrieved LST values. The RMSE of 0.9°C and 0.87°C and $R^2$ of 0.98 and 0.99 were obtained for 1998 and 2009 images, respectively. Land use types of the city of Yazd were identified and relationships between land use types, land surface temperature and NDVI index were analyzed. The Kappa coefficient and overall accuracy were calculated for accuracy assessment of the land use classification. The Kappa coefficient values are 0.96 and 0.95 and the overall accuracy values are 0.97 and 0.95 for 1998 and 2009 classified images, respectively. The results of this study showed that optical and thermal remote sensing methodologies can be used to research urban environmental parameters. Finally, it was found that special thermal conditions in Yazd were formed by land use changes. Increasing the area of asphalt roads, residential, commercial and industrial land use types and decreasing the area of the parks, green spaces and fallow lands types of land uses in Yazd caused a rise in surface temperature during the 11-year period. The results showed an increase of 1.45 degrees Celsius of the average surface temperature.

Keywords: Yazd, NDVI, Landsat Thematic Mapper, LST, Land use.

**1 Introduction**

A problem in the urban areas is surface temperature increasing due to conversion of vegetated surfaces into asphalt roads, residential, commercial and industrial areas. Nowadays, climate change in the cities is occurring by anthropogenic activities and land use changes. The atmospheric conditions of the urban areas, land surface temperature, warming, evaporation and absorption

of solar radiation may be changed by anthropogenic changes. The study of surface temperature in the cities located in the arid and semi-arid areas is necessary, because high temperature leads to energy consumption for cooling buildings, which is economically very costly especially in the warm months of year.

[revised manuscript text omitted]

**2.3 Calibration and validation of Landsat TM LST and land use classification**

The validation of Land Surface Temperature (LST) of satellite products is performed by using ground-based measurements of surface temperature. In the present study, ground-based data of land surface temperature received from Yazd Meteorological Bureau were used to validate LST of Landsat TM sensor. Measuring land surface temperature was performed using thermometers with SMT160 temperature sensor. The thermometers have a temperature range of -45 to 130 ° C. The most important characteristics of the SMT160 temperature sensor include absolute accuracy ∓0.7 ° C and measurement range 175 ° C. Geographical characteristics of the measurement points can be seen in Table 3.

Table 3

[revised manuscript text omitted]

**4 Discussions**

This paper proposed the TES algorithm to obtain LST from Landsat TM and change detection in land surface temperature in relation to land use change. The spatial scales of TIRS pixel (100 m), land use and NDVI products (30 m) can reduce accuracy of the results. The classified images show that the agricultural lands were classified as fallow land, and it also was observed that in some places of study area fallow lands have been combined with waste land and bare soil because of similar values of their spectral reflectance. As a result, the main factor of the conversion of land use types of study area in the 11-year period is the human activities for urban growth. Meanwhile, water class does not exist in the land use classification because of the lack of surface water resources in the study area. Based on the results, the classified images have significantly good accuracy over different land use types (Figures 4 and 5). Wheat and barley is mainly grown in the agricultural lands of the study area (seasonal plants) that in the performed classification are placed on the fallow land class because of similar values of spectral reflectance of these products and fallow lands. It should be noted that these agricultural products were harvested in May month. As mentioned in Section 3.2, the NDVI values decreased over the studied period of time (1998-2009) because of plant-covered surfaces reduction. It should be noted that central part of the study area had a slight decrease in the surface temperature. It was probably due to design and implementation of artificial green spaces and parks by managers in the city. Also, many cooling devices which are used in residential areas, especially in the summer were effective in creating this micro-climate. However, reduction of vegetation cover and NDVI values and the consequent rise in temperature were observed in the entire study area. A temperature higher of outlying parts than central part of city was due to keeping and increase parks and green spaces in the central part and the destruction of vegetation in the outlying parts of the city. Meanwhile, an increase the overall temperature of Yazd has different reasons, including vegetation loss and land use change in the area. It is generally observed that surface temperature has been increased in the all types of land use, but the greatest increase is registered in the commercial and industrial sites. It should be noted that residential areas had a slight decrease in the surface temperature. This is probably due to many cooling devices which are used in residential areas of the Yazd, especially in the summer. The high coefficient of determination ($R^2$) and low RMSE of satellite LST indicate that the study results have high accuracy. Values of NDVI index have an inverse relation with land surface temperature. This means that decrease in NDVI corresponded to an increase in temperature of land use types and vice versa. The strongest inverse relationship between surface temperature and NDVI values was observed in the fallow lands, parks and green spaces. By reducing vegetation density (in this study waste land and bare soil), inverse relation between surface temperature and NDVI value was weaker. By increasing vegetation cover, NDVI index values increased and surface temperature decreased. Also, an increase in temperature and a decrease in NDVI values were observed by increasing waste and barren lands in area. Land surface temperature can be estimated using a linear regression between surface temperature and NDVI, if NDVI values are known with reasonable accuracy. According to the received results, NDVI values were decreased during the study time. Land use types changes were causing surface temperature increasing, and temperature increasing subsequently was causing changes of NDVI and vegetation covers in the study area. The decrease in vegetation cover and an increase in residential areas are the main reasons of decreasing NDVI values of the Yazd. Land use gradual changes during the time are causing the temperature change. In the Yazd surface temperature has increased in the result of increases in asphalt roads, commercial, industrial and residential areas and a decrease in parks, green spaces and fallow land classes in 2009 compared to the year 1998. However, waste land and bare soil decreased, but mutually asphalt roads, commercial, industrial and residential areas increased, and these changes caused a rise in temperature. Other studies also confirm that the LST and NDVI changes are due to changes of the vegetation cover and residential areas (Gong Z. et al., 2015; Sandra E. et al., 2015; Valor E. et al., 1996; Wei L. et al., 2015 and Javed Mallick et al., 2008).

**5 Conclusions**

[revised manuscript text omitted]

06th Aug, 2009    Between Groups    0.295    5    0.059    0.000

Within Groups    0.000    24    0.000

Total    0.295    29